# Cricket-Enriched Oat Biscuit: Technological Analysis and Sensory Evaluation

**DOI:** 10.3390/foods9111561

**Published:** 2020-10-28

**Authors:** Barbara Biró, Mária Anna Sipos, Anikó Kovács, Katalin Badak-Kerti, Klára Pásztor-Huszár, Attila Gere

**Affiliations:** Institute of Food Technology, Faculty of Food Science, Szent István University, Villányi út 29-43, 1118 Budapest, Hungary; barbarabirophd@gmail.com (B.B.); siposa96@gmail.com (M.A.S.); Kovacs.Aniko94@szie.hu (A.K.); Badakne.Dr.Kerti.Katalin@szie.hu (K.B.-K.); Pasztorne.Huszar.Klara@szie.hu (K.P.-H.)

**Keywords:** entomophagy, edible insect, novel food, check-all-that-apply method, oat biscuit

## Abstract

Insect-containing products are gaining more space in the market. Bakery products are one of the most promising since the added ground insects can enhance not only the nutritional quality of the dough, but technological parameters and sensory properties of the final products. In the present research, different amounts of ground *Acheta domesticus* (house cricket) were used to produce oat biscuits. Colour, hardness, and total titratable acidity (TTA) values were measured as well as a consumer sensory test was completed using the check-all-that-apply (CATA) method. An estimation of nutrient composition of the samples revealed that, according to the European Union*’*s Regulation No. 1924/2006, the products with 10 and 15 g/100 g cricket enrichment (CP10 and CP15, respectively) can be labelled as protein sources. Results of the colour, TTA, and texture measurements showed that even small amounts of the cricket powder darkened the colour of the samples and increased their acidity, but did not influence the texture significantly. Among product-related check all that apply (CATA) attributes, fatty and cheesy flavour showed a significant positive effect on overall liking (OAL). On the other hand, burnt flavour and brown colour significantly decreased OAL. OAL values showed that consumers preferred the control product (CP0) and the product with 5 g/100 g cricket enrichment (CP5) samples over CP10 and rejected CP15.

## 1. Introduction

One of the most serious problems of the 21th century is that global food security of the growing population cannot be assured due to the limited freshwater resources and available cultivable land [1,2]. According to the Food and Agricultural Organization of the United Nations’ (FAO) 2019 report, the number of people suffering from hunger is about 820 million. Nutritional deficiencies mainly affects Africa, Southern and Western Asia, and Latin America [3]; however, other forms of malnutrition (overweight, obesity, and micronutrient deficiencies) are globally present, not just in the developing countries, but also in the developed countries. In total, about 2 billion people experience some level of food insecurity in the world. Consequently, it is necessary to focus not only on the supply of sufficient quantities of food, but also on the nutritional quality of the diet [4]. Dietary quality scores (DQSs) are used to evaluate the quality of diets. When DQSs are calculated, nutrients are often classified as qualifying or disqualifying, based on the dietary recommendations. According to these classifications, adequate intake of nutrients such as high-quality protein, dietary fibre, vitamin A, vitamin C, vitamin E, folate, calcium, and iron is considered to be qualifying. High saturated fat, added sugar, salt, and cholesterol content, as well as too low or too high energy intake, are classified as disqualifying. These factors are closely linked to undernutrition, obesity, and the development of non-communicable chronic diseases, such as cardiovascular diseases and type 2 diabetes mellitus [5,6].

According to several studies, the food industry is responsible for 26–50% of the greenhouse gas (GHG) emission, while agriculture is responsible for approximately 80% of the anthropological water footprint. High-quality protein production creates the greatest impact on the environment; since livestock breeding requires high amount of feed that is responsible for significant emission of GHGs, among others [7,8]. Based on FAO’s statement development on dietary guidelines, a healthy but also sustainable diet contains less food of animal origin (meat, fish, and dairy products) and legumes, wholegrain products, and seeds are instead recommended. The consumption of foods high in fat, salt, and sugar should be very limited; and efforts should also be made to consume minimally processed foods [9].

Several studies from the past decade have focused on the role of edible insects since they can be one of the possible solutions to address the problem of food security. Entomophagy or “eating insects” was common in the prehistoric era and is still a part of the diet of certain areas of Latin America, Africa, and Asia [10,11]. More than 2000 edible species are known, including beetles, caterpillars, wasps, ants, and grasshoppers [12]. The nutritional quality of edible insects is satisfying; however, it depends on the species, the feed, and the stage of the development [13]. Generally, they contain high-quality protein, and their fatty acid composition resembles those of poultry and fish, but with higher amount of polyunsaturated type [14]. Their carbohydrate composition is diverse; due to their exoskeletal chitin content, they are considered as a good source of dietary fibre. Regarding the micronutrients, their vitamin B and E, iron, magnesium, and zinc content are remarkable [12,14,15,16]. Thus, insects contain most of the nutrients considered to be qualifying, and could be used successfully in malnutrition management [17].

Numerous studies have shown that the farming of edible insects has a lower environmental impact than that of other livestocks’. The land use of insect farming is significantly lower, as insects are suitable for indoor, urban, and vertical farming. Many types of agricultural, industrial, and household wastes can be recycled as insect feed, since they can digest forage higher in dietary fibre. Their GHG and ammonia emissions are minimal compared to beef or pork breeding. The required water for producing a unit amount of insect protein is a fragment of the amount used for the production of other types of animal proteins [11,18].

Using insects as feed or food has been proven to be beneficial from both nutritional and environmental point of view. Commercial farming and insect-based food production require a strict regulatory system, which is still being developed in the European Union. According to the European Food Safety Authority’s (EFSA) 2015 scientific opinion, further data generation is required on the field of food safety, especially on microbiological and chemical hazards and allergenicity [19].

The consumer acceptance of insects as food depends on many influencing factors. Western people still have a negative attitude towards them; however, knowledge about edible insects and their nutritional value, previous taste experiences, and sensation seeking seem to increase acceptance. Many studies have proved that using insects in a non-visible form (e.g., insect powders) can increase the willingness to eat them. As a consequence, complementation of our everyday foods (e.g., bread, pasta, and other bakery and snack products), which are able to mask the insect component, can be the first step introducing insect-based foods to the market [20,21]. Several insect-containing food products have been described in the scientific literature recently, such as buckwheat pasta, breads, and energy bars [22,23,24,25].

Oat products are very popular nowadays since they are considered to be healthy. Oat *(Avena sativa)* is a cereal belonging to the *Poaceae* family. It has good nutritional qualities because of its relatively high protein, fibre, vitamin B, and mineral content [26]. The most important type of fibre in oats is the soluble β-glucan, which is a heterogeneous group of non-starch polysaccharides. Many studies have proved that this type of fibre has beneficial effects on diseases such as obesity, diabetes mellitus, hypertension, and dyslipidaemia, among others [27]. Shortbread-type biscuits are easy and simple to make, and many recipes are available that includes different type of cereal flours and also specifically oat flour.

The advantageous characteristics of insects and oat products as food ingredients can be mixed and a promising way of introducing insect-containing products would be the enrichment of oat biscuits with insect powders. Based on these, the aims of this study were to:present the usability of different amounts of insect powder in a biscuit product,examine the technological effect of insect enrichment on the finished products, anddiscover how the insect content of the products affects the overall liking (OAL), and which attributes are the drivers of liking.

## 2. Materials and Methods

### 2.1. Materials

Four biscuits were created and evaluated, oat and buckwheat flours served as their base. The proportion of buckwheat flour in each flour mixtures was 20 g/100 g, which was determined by carrying out and assessing pre-tests and pre-tastings. The purpose of the addition of the buckwheat flour was to refine the colour differences between the samples without significantly affecting the taste. House cricket (*Acheta domesticus*) powder was added in different amounts as shown in Table 1. Oat flour and buckwheat flour (Első Pesti Malom- és Sütőipari Zrt. Dunaharaszti, Hungary and Bonetta Bt; Lajosmizse, Hungary) were commercially purchased from Hungarian producers, *Acheta domesticus* powder was purchased from JR Unique Foods Ltd. (Udon Thani, Thailand). The other ingredients of the doughs were: unsalted butter, sour cream with 12% fat content (Alföldi Tej Kft; Székesfehérvár, Hungary), baking powder (Dr. Oetker Magyarország Élelmiszer Kft, Budapest, Hungary) and salt. Lactose-free versions of dairy products were used to avoid losing participants due to lactose intolerance. The compositions of each sample are shown in Table 1. Table 2 contains the nutritional composition of the ingredients.

Shortbread type biscuit samples were made in laboratory conditions, each in a separate container. For the proper homogenization, batches of 300 g dough were made. After weighing the oat and buckwheat flour and the cricket powder, the four flour mixtures were prepared. Next, the mixtures were mixed and homogenized by kneading and adding the weighed and cut butter, sour cream, baking powder and salt. The homogenization was performed with a Bosch MUM4830 food processor (Robert Bosch Kft; Budapest, Hungary). The prepared dough was rolled out to 3 mm thickness layer, then 5 cm diameter circular pieces were cut out of it. Finally, the biscuits were baked for 10 minutes in a Sveba Dahlen S300 mini batch oven (Sveba Dahlen, Fristad, Sweden) preheated to 180 °C, with the fan function on. The weight of the biscuits was 5.87 ± 1.24 g per piece. The baked biscuits are shown in Figure 1.

### 2.2. Methods

#### 2.2.1. Estimated Nutritional Value

Our calculations were based on the nutritional values labeled on the packaging of the used ingredients. The energy values were calculated according to the European Union*’*s Regulation No. 1169/2011 on the provision of food information to consumers. In this present calculation the used ingredients’ protein, carbohydrate, and fat content were considered [28]. During the development of the products we worked with a mixture of oat and buckwheat flours. In 100 g flour mixture we replaced 5, 10 and 15 g of the oat flour with cricket powder, while the amount of buckwheat flour in each flour mixture was 20 g*/*100 g, since our aim was to study the effects of substituting cereal flour with insect powder. Due to the addition of the other ingredients, the amount of cricket powder in 100 *g* of the final products were as follows: CP: 0 g, CP5: 3.24 g, CP10: 6.49 g, CP15: 9.73 g. Therefore, the products are referred to as “0, 5, 10, and 15 g*/*100 g house cricket containing flour mixture-based biscuits”.

#### 2.2.2. Technological Parameters and Quality Measurements

##### Colour Measurements

The colour of the baked samples was measured by using a Konica Minolta CR-310 Chroma Meter (Konica Minolta, Chiyoda, Tokyo, Japan). The colour parameters *L**, *a** and *b** are measures of lightness, redness/greenness and yellowness/blueness. The instrument was calibrated against a standard white tile (*L** = 97.63, *a** = 0.78 and *b** = 0.25). Since more biscuits of each batches were tested, the measurements were carried out in fourteen replicates.

##### Textural Hardness

Textural hardness was measured using a Stable Micro Systems TA.XT2i texture analyser (Stable Micro Systems, Godalming, United Kingdom) calibrated with a 2000 g load cell. For analysis, three-point bend rig was calibrated to a 10 mm height above the sample’s surface and programmed to approach at 1 mm·s^−1^. The samples were placed centrally over the supports. Upon contact with the surface, the probe pressed the samples at 3 mm·s^−1^ for 5 mm distance. The “resistance” of the samples against the moving probe was recorded. The measured maximum force value is referred to as the “hardness” of the sample. Measurements were carried out in fourteen replicates, since more biscuits of each batch were tested.

##### Total Titratable Acidity (TTA)

Total titratable acidity descirbes the total concentration of undissociated acids and free protons in the sample, that can react with a strong base. In bakery products, TTA is an indicator of microbial activity, since the moist dough is an adequate, nutrient rich media for acid-producing microorganisms [29].

Total titratable acidity was determined as described by Minervini, Lattanzi, De Angelis, Di Cagno, and Gobbetti [30]. 10 g of each biscuit sample was blended with 90 mL distilled water in a porcelain mortar. The suspensions were titrated to the final pH of 8.5 with 0.1 N NaOH. Changes of the pH value were followed by a HANNA Instruments Model pH 209 precision pH meter (HANNA Instruments, Woonsocket, RI, USA). The results are expressed as the amount of NaOH (ml) used. All measurements were carried out in triplicates.

#### 2.2.3. Consumer Sensory Analysis

100 consumers were invited to the test, 67 of whom evaluated the prepared four samples in a one-week session. The assessors were students of Szent István University, Hungary. Gender ratio were 35.82%/64.18% males/females. The age of the participants ranged between 18 and 35 years, 67.16% of them were 18 to 23 years old, 55.22% of them live in the capital, Budapest and 38.81% of them have already tried insects, or insect-based food.

To ensure the reliability of the results, consumers were instructed prior to the evaluation. All participants were informed about the insect content of the samples in both verbal and written forms. A declaration of volunteering was also filled by the assessors, as insects can be allergenic and commercialisation of insect-based products as food is currently not permitted in Hungary.

Recommendations of Kilcast were followed during the sample presentation [31]. According to the international practice, the biscuit samples were labelled with 3-digit random numbers and a balanced block design was also applied for the test [32]. Each assessor was given one piece of biscuit per sample. The average height was 3.30 ± 0.12 mm, the average diameter was 48.12 ± 0.97 mm, while the average weight was 6.13 ± 0.84 g per biscuit. RedJade^®^ software was used to conduct the test (RedJade Sensory Solutions, Martinez, CA, USA).

##### Check-All-That-Apply (CATA)

Check-all-that-apply questionnaires consist of multiple-choice questions. The participants are presented a list of phrases or words, of which they can select all, which they consider appropriate. Due to its different application possibilities, simplicity, fast execution and effectiveness, CATA experiments has become very popular in the last few years. Recently, the method is used for product sensory analysis with both trained panels and consumers, since the method is quick and easy, the task requires less cognitive effort from the assessor, and consumer-driven sensory characterization could be more useful for product optimization [33,34].

During the experiment, the assessors are asked to identify which sensory attribute from a given list is present in the sample [35]. The list of the attributes is usually built up from the sensory characteristics of the product, but can also contain hedonic terms, emotions, and non-sensory properties. The assessors can select without any constraint on the number of the given terms [33]. According to a 2013 study by Ares & Jaeger, the frequency of usage of the listed attributes can be increased by grouping them (e.g., flavour/taste related terms, odour related terms, etc.) and presenting them in a structural way from the appearance to the flavour [36].

The list of the used terms was compiled with a panel of 10 trained assessors on a consensus basis. The panel consisted of the trained individuals of the Sensory Laboratory, Szent István University, who are continuously trained according to the relevant ISO standard [37]. The panel received product specific training on insect-enriched food products by evaluations of prototype products. The terms are listed in Table 3.

The data table of CATA analysis contains the sensory attributes in the columns the assessors and products in the rows. This is a binary table, where 1 means that the attribute is identified by an assessor, while 0 means that the attribute is not perceived by the assessor in the sample.

During the presented study, consumers were also asked to evaluate overall liking (OAL) on a 9-point hedonic scale (1 = ‘‘dislike extremely”, 9 = ‘‘like extremely”).

#### 2.2.4. Data Analysis

Analysis of variance (ANOVA) and Tukey HSD *post hoc* tests were used to compare the means obtained during the technological evaluation and quality assessment of the samples, as well as to compare the consumer liking scores. CATA data analysis was done using multiple methods. Cohran*’*s Q test was used to test the independence of products and attributes. Correspondence analysis (CA) was applied to visualize and interpret the products/attributes cross table, where the obtained inertia value indicates the quality of the analysis. Generally, higher inertia means higher quality analysis. Since our analysis contained overall liking assessment of the samples, principal coordinate analysis (PCoA) was run on the CATA and overall liking data. Since CATA variables and overall liking are measured on different scales (CATA—binary scale, overall liking—9-point category scale), they need to be transformed first. PCoA computes the chi-squared distance matrix of the CATA and overall liking variables, then centres the distance matrix by rows and columns, finally decomposes the eigenvalues of the centred distance matrix. This way PCoA enables us to visualize the CATA and overall liking variables on one plot where the distances among the variables express their similarities/dissimilarities [38].

In order to assess the attributes’ impact on overall liking, mean drop analysis was also conducted. During mean drop analysis, the mean overall liking is computed for each attribute when the attribute is present (OALpres) and absent (OALabs). OALabs is deducted from OALpres which gives the mean impact of the attribute. If the mean impact is lower than 0, the attribute has a negative impact on overall liking. Similarly, a positive mean impact value indicates positive effect.

ANOVA, Cohran*’*s Q test, CA, and PCoA were calculated using XL-Stat ver. 2019.2.2 (Addinsoft, 2019).

## 3. Results

### 3.1. Nutritional Values

Our calculations were based on the nutritional data labeled on the packaging of the used ingredients. Adding house cricket powder to the flour mixture increased the protein content from 9.48 g/100 g to 11.22 g/100 g in the case of 5 g/100 g enrichment (CP5), to 12.97 g/100 g at 10 g/100 g enrichment (CP10), and to 14.71 g/100 g at 15 g/100 g enrichment (CP15). Carbohydrate content was decreased from 38.27 g/100 g to 36.67 g/100 g in the case of CP5, 35.06 g/100 g in the case of CP10, and to 33.46 g/100 g in the case of CP15. The energy value was elevated 8.46 kcal, in the case of 15 g/100 g enrichment. The fat content shows a smaller difference (increased from 23.69 g/100 g to 24.68 in the case of CP15), while the fibre content decreased in proportion to the enrichment (decreased from 7.00 g/100 g to 5.98 g/100 g in the case of CP15) (Table 4).

Similar results were observed in the case of the protein content of wheat flour breads with house cricket and cinereous cockroach powder enrichment [24,25], rice flour cakes with Bombay locust powder enrichment [39], and buckwheat pasta with silkworm enrichment [23].

The reason of the decrease in fibre content is the decreased amount of oat flour, which has higher fibre content, than house cricket powder. The slightly increased fat content of the enriched samples is derived from the higher fat content of the insect powder. Similar results of change in fat content were presented in other studies, however, the fibre content of bread and snacks showed an increasing tendency in proportion to the insect enrichment [40,41]. This may be the consequence of the lower fibre content of the wheat flour, which is used as a control in other studies.

The European Union’s Regulation No. 1924/2006 on nutrition and health claims made on foods states: ‘a claim that a food is a source of protein, and any claim likely to have the same meaning for the consumer, may only be made where at least 12% of the energy value of the food is provided by protein’ [42]. According to this Regulation, the products with higher enrichment can be labelled as protein source, since 12.77% (CP10) and 14.39% (CP15) of their energy value is provided by protein (Table 4).

### 3.2. Technological Parameters and Quality Measurements

#### 3.2.1. Colour Measurements

Results of the colour measurement showed that even small amounts of insect enrichment influence the colour of the samples, as all colour measurement values changed proportionally with the amount of cricket powder. The *L** value of the control sample (CP0) was 63.50 ± 1.77, while CP15 showed 50.08 ± 0.73, indicating darkening effect. The *a** value increased from 7.92 ± 0.46 (CP0) to 9.53 ± 0.43 (CP15), which means that more redness appeared in the colour of the samples. Our results are in line with literature data, as the *L** value is also decreased and *a** value is also increased in the case of cockroach enriched bread products and locust enriched rice cakes [24,39]. Similarly to *L**, *b** values were also decreased: CP0 showed 25.47 ± 0.61, while CP15*′*s value was 22.67 ± 0.76, which suggests that the samples became less yellow. According to earlier studies, the *b** value shows an increasing tendency in insect-enriched bakery goods [40]. Our results may differ since the base of the biscuit samples were oat and buckwheat flour. Consequently, the *L**, *a** and *b** values of the control sample were higher than the wheat flour-based products of the cited studies. Significant difference was found among *L** (F(3,52) = 335.722, *p* < 0.0001), *a** (F(3,52) = 42.498, *p* < 0.0001) and *b** (F(3,52) = 73.224, *p* < 0.0001) values of all samples. The results of the colour measurements are listed in Table 5.

#### 3.2.2. Hardness

According to the results of hardness, the added amounts of cricket powder do not significantly influence the hardness of the samples (F(3,52) = 0.887, *p* = 0.454), however, an increasing tendency of these values is observable in proportion to the cricket powder content. The hardness (maximum force value of the resistance of the pressed probe) of the control sample (CP0) was 160.21 ± 43.90 N. In the case of CP5, CP10 and CP15 the values were 164.62 ± 44.16 N, 185.03 ± 69.48 N, and 183.72 ± 40.79 N, respectively. Other studies have also found that textural hardness of bread products increased with the addition of insect powder [40,43], but showed a decreasing tendency in the case of rice-flour cakes [39].

#### 3.2.3. Total Titratable Acidity (TTA)

TTA showed an increasing tendency in proportion to the amount of house cricket powder in the samples. Significant differences were found among the TTA of the four samples (F(3,4) = 187.763, *p* < 0.01). The acidity of the control sample (CP0) was 9.95 ± 0.35, which increased to 13 ± 0.14 in CP5. In the case of CP10 and CP15, TTA were 15 ± 0.42 and 17.65 ± 0.35, respectively. Similar results were presented in other studies on snacks enriched with lesser mealworm and breads enriched with house cricket [25,41].

### 3.3. Consumer Sensory Analysis

#### 3.3.1. Liking Variables

One-way analysis of variance of the liking variables revealed that significant difference exists among samples for colour, flavour, and overall liking (OAL). The addition of cricket powder significantly changed the colour of the samples, making them darker, redder and less yellow, which was confirmed by the results of colour liking values. Colour of CP5 received the highest liking values, which was not significantly different from that of CP0. Flavour of CP0 and CP5 were rated similarly, indicating that the 5 g/100 g substitution of cricket powder has no effect on the consumer*’*s opinion. On the other hand, CP10 and CP15 received low flavour liking values. Regarding overall liking, consumers preferred CP0 and CP5 and clearly rejected CP15. Odour and texture liking were not affected by the addition of different amounts of cricket powder, the latter is supported by the fact that no significant differences were found among the hardness values of three samples. Other studies showed corresponding results. In the case of biscuits fortified with edible termites, the control sample received the highest overall acceptability, and the panel preferred the 5% insect-containing biscuit among the enriched products. Also, no significant differences were found among the liking variables of the texture. Among the colour of the products, the colour of the sample with 5% insect content received the highest liking values, as well as in our study [44].

10% enrichment was preferred in the case of buckwheat-pasta enriched with silkworm powder, bread enriched with grasshopper powder and bread enriched with cricket powder. Furthermore, bread enriched with 10% flour from cinereous cockroach showed no significant differences from the control [24,25,43]. The results of the consumer sensory analysis are shown in Table 6.

#### 3.3.2. Check-all-that-apply (CATA)

CATA questionnaire consisted of 38 terms, of which the ten most frequently marked were Crumbly, Friable, Pleasant odour, Long lasting taste, Just-about-right colour, Brown colour, Sticky, Salty taste, Dry and Seedy flavour. The least used five terms were Hard, Spicy flavour, Fishy odour, Sweet taste and Fishy flavour. In order to get more accurate results, the attributes that do not differentiate the samples significantly were filtered out based on the Cochran’s Q test. Henceforth, only the remaining 25 significant properties will be used in the analysis. Table 7 shows the frequencies of marking of these attributes in the case of all samples.

According to the correspondence analysis (Figure 2a), the assessors associated different attributes with each sample. CP0 was divisive, as the consumers marked that the flavour and odour of the sample were also not good enough and just-about-right. CP5 was the most liked insect enriched biscuit, which is reflected in this analysis, as *Just-about-right colour*, *Tasty* and *Pleasant odour* attributes are close to the sample, as well as *Soft* and *Fatty* textures. *Grainy appearance* and *Granular texture*, *Brown colour*, *Toasty odour* and *Long lasting taste* appear along with samples CP10 and CP15. Negative properties such as *Burnt odour*, *Too dark colour*, *Too strong taste* and *Too strong odour* were more associated with these samples. Among the animal notes, *Cheese flavour* was mostly marked in the case of CP5, while *Fishy odour* and *Fishy flavour* in the case of CP10. *Hard texture* and *Earthy odour* were chosen by the assessors to describe CP15.

Since principal coordinate analysis (Figure 2b) visualizes the overall liking data (OAL) and the marked CATA terms together, drivers of liking can be easily defined. Hedonic terms with positive meaning *(Tasty*, *Just-about-right colour*, *Pleasant odour)* are close to OAL, such as *Toasty odour*, *Friable* texture, and *Cheesy flavour*. Some of the negative meaning hedonic terms *(Too weak odour*, *Too light*, *Too weak flavour)* are also close, which means that the assessors gave higher OAL scores when some properties were “not enough”, than when they were “too much”. *Hard* and *Granular* textures are on the opposite side of OAL, just as *Too dark* colour and *Brown colour*, *Too strong odour*, *Earthy* and *Fishy odour*, *Burnt* and *Fishy flavour*, which means that these attributes were less liked in the products. It can be observed that attributes close to OAL were, according to the correspondence analysis, characteristics of the CP0 and CP5 samples, while the more distant ones were more typical in the case of CP10 and CP15 samples. This confirms the result of the analysis of the liking variables, which showed that CP0 and CP5 were more liked.

Figure 2c presents the results of mean drop analysis of the CATA attributes. Highest mean impact was observed in the case of *Tasty*, however, only 23% of consumers marked as present. *Friable* was rated more than 68% of the consumers, which means strong consensus. Among product-related attributes, *Fatty* and *Cheesy flavour* showed significant positive effect on overall liking. On the other side, *Burnt flavour* and *Brown colour* showed significant negative impact on overall liking.

Figure 3 presents separate PCoAs of the four samples. *Too light* attribute is only marked in the case of CP0 (Figure 3a), while *Just-about-right colour* is located near to OAL in the case of CP5 (Figure 3b) and CP10 (Figure 3c). These suggest, that slightly darker colour enhances, while too dark colour decreases OAL. *Tasty* attribute is considered as a driver of liking, since it goes along OAL in every case. However, CP15 (Figure 3d) shows that higher amount of insect enrichment pushes *Tasty* further from OAL. *Sweet taste* is located close to OAL when no enrichment is done, while OAL of CP5 and CP10 is less influenced by the attribute and no participant marked *Sweet taste* while testing CP15. *Too dark* colour and *Burnt flavour* are far away from OAL, meaning these attributes have a decreasing effect on OAL.

Literature data is very limited on sensory evaluation of insect-enriched products performed with check-all-that-apply method. Mealworm-containing meatballs and dairy drink products were tested with CATA analysis; however, the attributes of these products are difficult to compare with the biscuits’ we developed. Nevertheless, with the addition of ground insects, a few similar properties appeared as in the case of fortified biscuits, e.g., grainy, sticky, and dry [45].

Comparing our study to the international literature, this is the first research which used oat and buckwheat flour as a base of cricket enriched biscuits. These flours have better nutritional characteristics in terms of high protein, higher dietary fibre, vitamin B and mineral content. In the case of functional foods, the acceptance of consumers is higher if the products are considered healthy [46]. However, our products cannot be considered as functional foods, consumer behavior might be similar. Nevertheless, oat and buckwheat can serve as bases of gluten free products. Our results support the evidence that pairing insects with these flours is viable options in order to develop novel gluten free products that could gain the acceptance of consumers [47,48].

From the methodological point of view, there is no existing study, which used check-all-that-apply analysis on ground-insect enriched products. As a result, our study provides a set of CATA descriptors, which can be applied in future studies.

## 4. Conclusions

The obtained results suggest that samples containing 10 g/100 g flour mixture (CP10) and 15 g/100 g flour mixture (CP15) *Acheta domesticus* powder can be labelled as protein source based on the corresponding EU regulation. However, consumer sensory analysis revealed that CP10 and CP15 were significantly less liked compared with the control and 5 g insect/100 g flour mixture (CP5) based sample. The rejection can be attributed to the changes in appearance and not due to changes of textural attributes, as the technological analysis suggested. The major factors of rejection were *Brown colour* and *Burnt flavour*; hence further product developments should address these issues.

Literature data suggest that consumers prefer insect containing products at different levels, however, there are limited results available about biscuits, since the majority of the publications have focused on other bakery products.

Our results raise the attention of policy makers and producers to the fact that insects enhance the nutritional quality of bakery products even if they are made from gluten free cereals and/or pseudocereals.

Limitations of our study are the lack of representative sampling; however, it is still in line with Næs’ recommendation [49]. According to a 2017 study, Hungarian consumers show slight rejection to insects as food, therefore, these results should not be generalized [50].

Further analysis should be carried out to test the effect of different species on the sensory attributes of insect enriched bakery products. Sensory attributes–therefore acceptance–might also be influenced by different base materials (e.g., flour types and mixtures), spices (salted or sweet products) and processing technologies (e.g., drying, frying, cooking, baking).

## Figures and Tables

**Figure 1 foods-09-01561-f001:**
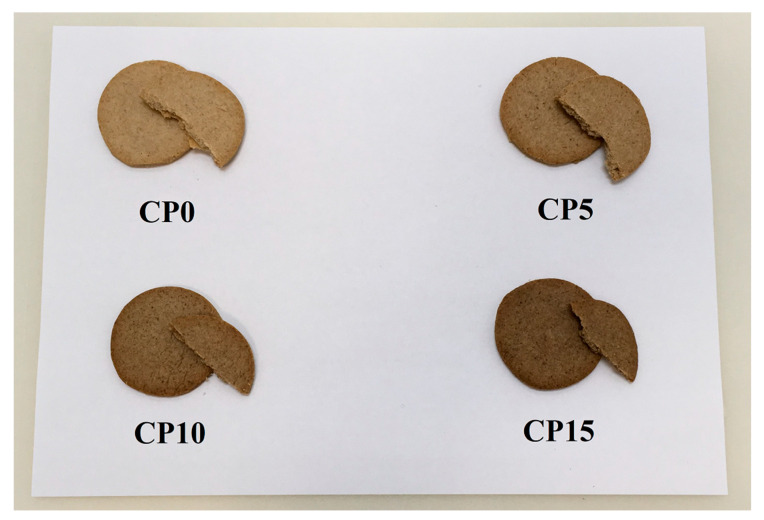
The prepared biscuits. CP0—0 g/100 g house cricket containing flour mixture-based biscuit, CP5—5 g/100 g house cricket containing flour mixture-based biscuit, CP10—10 g/100 g house cricket containing flour mixture-based biscuit, CP15—15 g/100 g house cricket containing flour mixture-based biscuit.

**Figure 2 foods-09-01561-f002:**
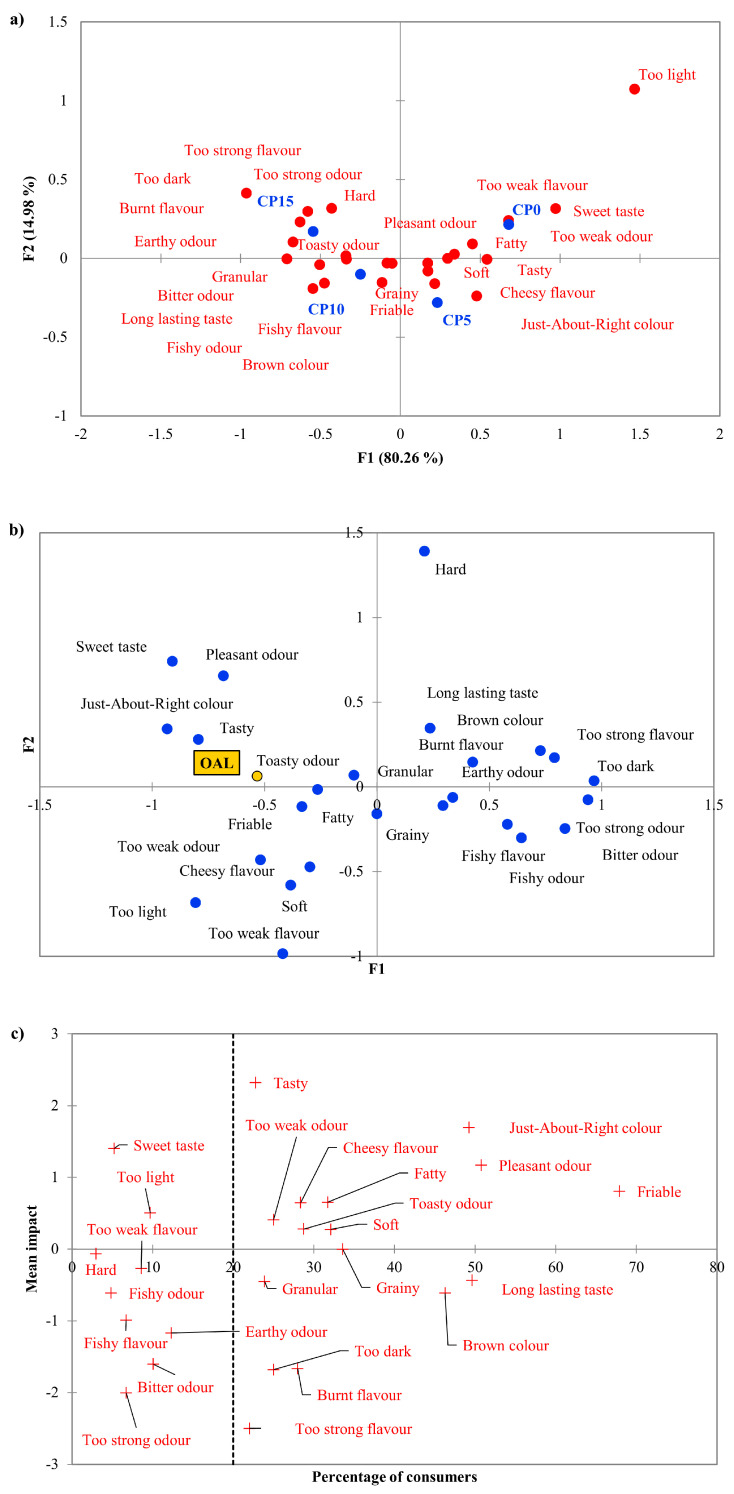
Visualized results of the check-all-that-apply (CATA) analysis of the four biscuit samples. (**a**) Correspondence analysis of the used CATA terms and the four samples. (**b**) Principal Coordinate Analysis of the used CATA terms and Overall Liking (OAL) scores. (**c**) The mean impact of the marked attributes on overall liking, visualized with the percentage of consumer who marked those attributes (dashed line). CP0—0 g/100 g house cricket containing flour mixture-based biscuit, CP5—5 g/100 g house cricket containing flour mixture-based biscuit, CP10—10 g/100 g house cricket containing flour mixture-based biscuit, CP15—15 g/100 g house cricket containing flour mixture-based biscuit. OAL stands for overall liking. Only those 25 CATA terms are shown that significantly differentiate the four samples.

**Figure 3 foods-09-01561-f003:**
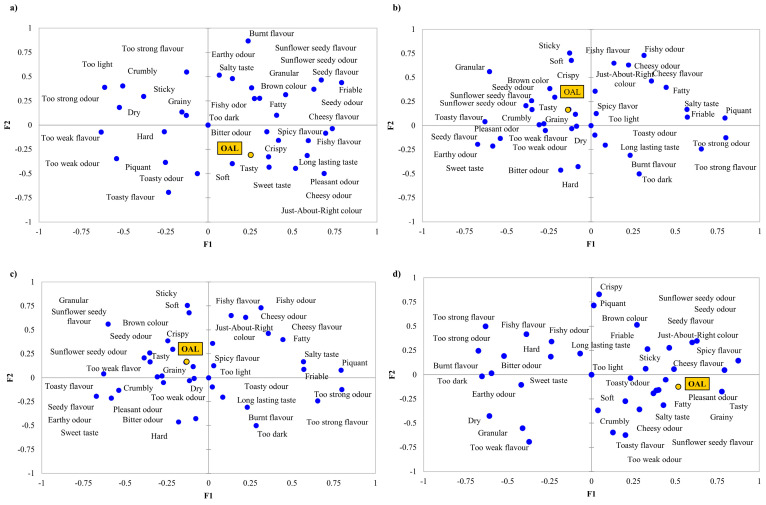
Principal coordinate analysis (PCoA) plots of the four samples. (**a**) CP0—0 g/100 g house cricket containing flour mixture-based biscuit, (**b**) CP5—5 g/100 g house cricket containing flour mixture-based biscuit, (**c**) CP10—10 g/100 g house cricket containing flour mixture-based biscuit, (**d**) CP15—15 g/100 g house cricket containing flour mixture-based biscuit. OAL stands for overall liking.

**Table 1 foods-09-01561-t001:** The composition of the evaluated biscuits. The amount of ingredients is expressed per 100 g of flour mixtures.

Sample	*Acheta domesticus* * Powder (g)	Oat Flour (g)	Buckwheat Flour (g)	Butter (g)	Sour Cream (g)	Baking Powder (g)	Salt (g)
CP0	0	80	20	33.9	20.3	0.4	0.7
CP5	5	75	20	33.9	20.3	0.4	0.7
CP10	10	70	20	33.9	20.3	0.4	0.7
CP15	15	65	20	33.9	20.3	0.4	0.7

* house cricket, CP0—0 g/100 g house cricket containing flour mixture-based biscuit, CP5—5 g/100 g house cricket containing flour mixture-based biscuit, CP10—10 g/100 g house cricket containing flour mixture-based biscuit, CP15—15 g/100 g house cricket containing flour mixture-based biscuit.

**Table 2 foods-09-01561-t002:** The energy value, macronutrient, and fibre composition of the used ingredients. Only the quantitatively relevant ingredients are listed. The nutritional values are given in their density (g or kcal) in 100 g of each ingredient.

Ingredient (100 g)	Energy (kcal)	Protein (g)	Carbohydrates (g)	Fat (g)	Fibre (g)
Cricket powder	457	67.8	5.5	18.2	0.5
Oat flour	370	14.0	55.0	8.0	11.0
Buckwheat flour	334	12.6	70.6	1.0	10.0
Lactose free butter	717	0.8	0.6	81.1	0.0
Lactose free sour cream (12% fat)	135	3.0	3.4	12.0	0.0

**Table 3 foods-09-01561-t003:** Attributes evaluated by consumers using CATA questions for sensory characterization of cricket enriched oat biscuits.

Product Property	CATA Terms
Appearance	too dark, too light, just-about-right colour, brown colour, grainy
Odour	too strong odour, too weak odour, cheesy odour, bitter odour, seedy odour, earthy odour, sunflower-seedy odour, toasty odour, pleasant odour, fishy odour
Texture	friable, hard, soft, crumbly, fatty, crispy, granular, dry, sticky
Flavour	too strong flavour, too weak flavour, cheesy flavour, seedy flavour, spicy flavour, salty taste, sunflower-seedy flavour, toasty flavour, tasty, sweet taste, piquant, fishy flavour, burnt flavour, long lasting taste

**Table 4 foods-09-01561-t004:** Biscuit compositions and calculated nutritional values. Amounts of each material are presented next to the name of the material. Nutritional values correspond to the quantity each sample contains. The table only lists the ingredients which contain energy-providing nutrients.

Sample	Ingredient	Amount (g/100 g)	Energy (kcal/100 g)	Protein (g/100 g)	Carbohydrates (g/100 g)	Fat (g/100 g)	Fibre (g/100 g)	Protein/Energy Value (%)
**CP0**	Cricket powder	0.00	0.00	0.00	0.00	0.00	0.00	9.46
	Oat flour	51.88	191.96	7.26	28.53	4.15	5.71	
	Buckwheat flour	12.97	43.32	1.63	9.16	0.13	1.30	
	Butter	21.98	157.63	0.19	0.13	17.83	0.00	
	Sour cream	13.16	17.77	0.39	0.45	1.58	0.00	
	Overall	100.00	410.68	9.48	38.27	23.69	7.00	
**CP5**	Cricket powder	3.24	14.82	2.20	0.18	0.59	0.02	11.13
	Oat flour	48.64	179.96	6.81	26.75	3.89	5.35	
	Buckwheat flour	12.97	43.32	1.63	9.16	0.13	1.30	
	Butter	21.98	157.63	0.19	0.13	17.83	0.00	
	Sour cream	13.16	17.77	0.39	0.45	1.58	0.00	
	Overall	100.00	413.50	11.22	36.67	24.02	6.66	
**CP10**	Cricket powder	6.49	29.64	4.40	0.36	1.18	0.03	12.77
	Oat flour	45.40	167.96	6.36	24.97	3.63	4.99	
	Buckwheat flour	12.97	43.32	1.63	9.16	0.13	1.30	
	Butter	21.98	157.63	0.19	0.13	17.83	0.00	
	Sour cream	13.16	17.77	0.39	0.45	1.58	0.00	
	Overall	100.00	416.32	12.97	35.06	24.35	6.32	
**CP15**	Cricket powder	9.73	44.46	6.60	0.54	1.77	0.05	14.39
	Oat flour	42.15	155.97	5.90	23.18	3.37	4.64	
	Buckwheat flour	12.97	43.32	1.63	9.16	0.13	1.30	
	Butter	21.98	157.63	0.19	0.13	17.83	0.00	
	Sour cream	13.16	17.77	0.39	0.45	1.58	0.00	
	Overall	100.00	419.14	14.71	33.46	24.68	5.98	

CP0—0 g/100 g house cricket containing flour mixture-based biscuit, CP5—5 g/100 g house cricket containing flour mixture-based biscuit, CP10—10 g/100 g house cricket containing flour mixture-based biscuit, CP15—15 g/100 g house cricket containing flour mixture-based biscuit.

**Table 5 foods-09-01561-t005:** Means and standard deviations of the colour measurement parameters across samples.

Sample	*L**	*a**	*b**
CP0	63.50 ± 1.77 ^a^	7.92 ± 0.46 ^a^	25.47 ± 0.61 ^a^
CP5	58.24 ± 0.61 ^b^	8.66 ± 0.21 ^b^	24.88 ± 0.23 ^b^
CP10	53.74 ± 1.23 ^c^	9.08 ± 0.40 ^c^	23.76 ± 0.40 ^c^
CP15	50.08 ± 0.73 ^d^	9.53 ± 0.43 ^d^	22.67 ± 0.76 ^d^

*L** value stands for the lightness from black (0) to white (100), *a** value stands for from green (−) to red (+), and *b** value stands for from blue (−) to yellow (+). Superscript letters denote homogenous subgroups defined by Tukey HSD *post hoc* test. OAL denotes overall liking. CP0—0 g/100 g house cricket containing flour mixture-based biscuit, CP5—5 g/100 g house cricket containing flour mixture-based biscuit, CP10—10 g/100 g house cricket containing flour mixture-based biscuit, CP15—15 g/100 g house cricket containing flour mixture-based biscuit.

**Table 6 foods-09-01561-t006:** Means and standard deviations of the liking variables across samples.

Sample	Colour	Odour	Texture	Flavour	OAL
CP0	**7.03 ± 1.78 ^ab^**	6.46 ± 1.58 ^a^	5.93 ± 1.74 ^a^	**6.55 ± 1.86 ^a^**	**6.57 ± 1.71 ^a^**
CP5	**7.48 ± 1.43 ^a^**	7.09 ± 1.53 ^a^	5.87 ± 1.97 ^a^	**6.27 ± 2.12 ^a^**	**6.42 ± 1.88 ^ab^**
CP10	**6.42 ± 1.68 ^b^**	6.55 ± 1.71 ^a^	5.93 ± 1.72 ^a^	**5.21 ± 2.12 ^b^**	**5.49 ± 1.94 ^bc^**
CP15	**5.33 ± 2.00 ^c^**	6.10 ± 1.93 ^a^	5.54 ± 1.92 ^a^	**4.70 ± 2.40 ^b^**	**4.78 ± 2.04 ^c^**

Bold indicates significant differences among samples and/or clusters defined by analysis of variance (*p* < 0.05). Superscript letters denote homogenous subgroups defined by Tukey HSD *post hoc* test. OAL denotes overall liking. CP0—0 g/100 g house cricket containing flour mixture-based biscuit, CP5—5 g/100 g house cricket containing flour mixture-based biscuit, CP10—10 g/100 g house cricket containing flour mixture-based biscuit, CP15—15 g/100 g house cricket containing flour mixture-based biscuit.

**Table 7 foods-09-01561-t007:** Frequency of marking of the CATA terms for all four biscuit samples.

Attribute	Marked as Present	Attribute	Marked as Present
Crumbly	196	**Too dark**	67
**Friable**	182	**Too weak odour**	67
**Pleasant odour**	136	Sunflower seedy odour	65
**Long lasting taste**	133	**Granular**	64
**Just-about-right colour**	132	**Tasty**	61
**Brown colour**	124	**Too strong flavour**	59
Sticky	123	Sunflower seedy flavour	59
Salty taste	115	Crispy	36
Dry	104	**Earthy odour**	33
Seedy flavour	94	**Bitter odour**	27
Cheesy odour	92	**Too light**	26
**Grainy**	90	**Too weak flavour**	23
Seedy odour	86	Piquant	20
**Soft**	86	**Too strong odour**	18
**Fatty**	85	**Fishy flavour**	18
**Toasty odour**	77	**Sweet taste**	14
**Cheesy flavour**	76	**Fishy odour**	13
**Burnt flavour**	75	Spicy flavour	11
Toasty flavour	68	**Hard**	8

Attributes that differentiate the samples significantly are highlighted in bold (Cochran’s Q test).

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
