# Peer review of "Cricket-Enriched Oat Biscuit: Technological Analysis and Sensory Evaluation"

_foods, 2020, doi:10.3390/foods9111561_

Round 1

Reviewer 1 Report

The objective of the work is to assess how consumers perceive certain attributes related to the consumption of insect-based food.
In general the work is well structured. For example, the Introduction is clear in the objectives and in the representation of the scientific context.
The description of the method is clear even if the sample of consumers interviewed (67 students) leaves some doubts about the external validity of the results.
Finally, the results are clearly presented.
My main concern on this paper concerns what is the new contribution of this research compared to the current scientific literature on the subject. Perhaps, the Authors should foresee a new section of Discussion that describes the novelty aspects of this research with respect to the existing literature. This would help the reader to understand how this research contributes to the topic of insect food consumption.
Moreover, in the Conclusions section, the Authors should better describe the implications of their findings. For example, what implications emerge for companies or policy makers?
Finally, the Authors should better highlight the limitations of their study, for example the small size of the interviewed sample.

Author Response

Reviewer 1

The objective of the work is to assess how consumers perceive certain attributes related to the consumption of insect-based food.
In general the work is well structured. For example, the Introduction is clear in the objectives and in the representation of the scientific context.

Thank you for your positive comment!

The description of the method is clear even if the sample of consumers interviewed (67 students) leaves some doubts about the external validity of the results.

According to the international literature of consumer sensory analysis, the recommended number of assessors is around 100, however, Næs recommends a minimum of 60 participants (Næs, Brockhoff, & Tomic, 2010). Over a hundred people were invited to the sensory evaluation, but unfortunately not all invited participants came, in addition, we had to exclude the data of some participants from the analyses. Thus, we were able to work with data from a total of 67 assessors, which, however, is in line with Næs’ recommendation.

Næs, T., Brockhoff, P. B., & Tomic, O. (2010). Statistics for Sensory and Consumer Science. Hoboken: Wiley.

Finally, the results are clearly presented.
My main concern on this paper concerns what is the new contribution of this research compared to the current scientific literature on the subject. Perhaps, the Authors should foresee a new section of Discussion that describes the novelty aspects of this research with respect to the existing literature. This would help the reader to understand how this research contributes to the topic of insect food consumption.

Comparing our study to the international literature, this is the first research which used oat and buckwheat flour as a base of cricket enriched biscuits. Besides the development of a novel food product, most of the existing researches use wheat flour as a base. Oat and buckwheat flour have better nutritional characteristics in terms of high protein, higher dietary fiber, vitamin B and mineral content. In the case of functional foods, consumer acceptance is higher, when the product is considered healthy (Verbeke, 2005). However, our products can not be considered as functional foods, consumer behavior might be similar. Nevertheless, oat and buckwheat can serve as bases of gluten free products. With our work it is proven that pairing insects with these is viable option for the development of novel gluten free products, which are accepted by consumers (Nisen et al., 2020, Mancini et al., 2020).

Nissen, L., Samaei, S. P., Babini, E., & Gianotti, A. (2020). Gluten free sourdough bread enriched with cricket flour for protein fortification: Antioxidant improvement and Volatilome characterization. Food Chemistry, 333(May), 127410. https://doi.org/10.1016/j.foodchem.2020.127410

Mancini, S., Fratini, F., Tuccinardi, T., Degl’Innocenti, C., & Paci, G. (2020). Tenebrio molitor reared on different substrates: is it gluten free? Food Control, 110(November 2019), 20–23. https://doi.org/10.1016/j.foodcont.2019.107014

Moreover, in the Conclusions section, the Authors should better describe the implications of their findings. For example, what implications emerge for companies or policy makers?
Finally, the Authors should better highlight the limitations of their study, for example the small size of the interviewed sample.

Thank you for your comment. We included the following sentences into the conclusion section in order to highlight the implications of our findings and to raise the readers’ attention to the limitations of the study:

Our results raise the attention of policy makers and producers to the fact that insects enhance the nutritional quality of bakery products even if they are made from gluten free cereals and/or pseudocereals.

Limitations of our study are the lack of representative sampling; however, it is still in line with Næs’ recommendation. According to a 2017 study, Hungarian consumers show slight rejection to insects as food therefore, these results should not be generalized (Gere et al., 2017).

Gere, A., Székely, G., Kovács, S., Kókai, Z., & Sipos, L. (2017). Readiness to adopt insects in Hungary: A case study. Food Quality and Preference, 59. https://doi.org/10.1016/j.foodqual.2017.02.005

Reviewer 2 Report

Cricket enriched oat biscuit: technological analysis and sensory evaluation
By:  Barbara Biró et al.

This is an overall excellent study and analysis: well planned, well executed and well analysed. I have only a few remarks that would help to make the paper even more complete. Although generally written in very good English, there are the odd grammatic and vocabulary mistakes, which should be corrected in the revision, but which I shall not specifically mark in my review.

Line 12: to write about “…technological and sensory properties of the final products” is a combination between two descriptors that in this context do not fit well. Yes, “sensory properties” is fine, but “technological properties of the final products” is not; at least it is not clear to this reviewer how the final products can have ‘technological properties’.

Line 13: “Different amounts….were used”  (not was)

Line 19: do not use “however” here, but write “but do not influence…”

Line 29: after “land [1]” you ought to cite the pioneering paper by Meyer-Rochow 1975, which started the ball rolling and led to all this current interest in insects as food (“Can Insects Help To Ease The Problem Of World Food Shortage?” Search 6(7), 261-262).

Line 31: what are subregions of Africa?

Lines 30 and 32: I suggest you first write “Malnutrition…” instead of ‘undernutrition’ and on line 32 continue with “other forms of food-related problems (hunger….”) or  “additional nourishment problems such as hunger….”

Lines 35-37: very good!

Line 40: write “…or too high an energy intake.”

Line 44: “…studies, the food industry…”

First paragraph: very good!

Line 61: Check out the paper by Chakravorty et al (Int. J. Vitam. Nutr. Res., 81 (1), 2011, 1 – 14). It may be worth referring to it.

Line 71: Delete ‘However’ (you tend to use ‘however’ too often) and start “Using insects as….points of view, but industrial farming…”  I wonder if ‘commercial farming’ would not be more appropriate than ‘industrial’?

Line 76: “Western people are still disgusted…”  I think you cannot make such a sweeping statement.  Not all “Western people” are disgusted; there are many reasons why not many Westerners enjoy consuming insects and disgust is only one of several factors. Using that word is insulting  to millions of people who love insects (not necessarily to eat), but to enjoy, to keep as pets, to study, etc.

Line 118: “at once” seems hardly the correct expression. What do you exactly mean to say?

Line 138 and Line 171: I recently reviewed a paper in which ‘spreadability’ (how quickly and fast the dough spread) was shown to depend on the amount of insect flour in the dough and how this affected the thickness and size of the cookie. Was there no such effect in your study?

Line 159-163: The educational backgrounds of the ‘assessors’ were more or less the same (all students). If you were to predict (by extension) how the general population of Budapest might have reacted, could you speculate in Discussion what the outcome may have been?

Table 2, Use British spelling “Odour” and ”Flavour”

Page 7, first paragraph: excellent

Figures: all necessary and acceptable

Line 392: write “…compared with the control…”  When you compare something ‘t’, you are expressing likeness, but if the task was a true comparison, it has to be ‘with’ (Example: you can compare Italy’s weather to that of New Zealand  = similar. Or you can compare Italy’s weather with that of New Zealand = seeking a true comparison).

The authors may want to read and cite this interesting paper “Factors Predicting the Intention of Eating an Insect-Based Product” by Simone Mancini ,Giovanni Sogari ,Davide Menozzi ,Roberta Nuvoloni ,Beatrice Torracca ,Roberta Moruzzo and Gisella Paci: Foods 2019, 8(7), 270; https://doi.org/10.3390/foods8070270 - 19 Jul 2019, in which the authors write: “Sensory scores highlighted that participants gave “insect-labelled” samples higher scores for flavour, texture, and overall liking; nevertheless, participants indicated that they were less likely to use the “insect-labelled” bread in the future.”

Author Response

Reviewer 2

This is an overall excellent study and analysis: well planned, well executed and well analysed. I have only a few remarks that would help to make the paper even more complete.

Thank you for your kind comment. We did our best to make the paper better and addressed all the issues the reviewers raised.

Although generally written in very good English, there are the odd grammatic and vocabulary mistakes, which should be corrected in the revision, but which I shall not specifically mark in my review.

The paper has been sent for proofreading. As many small language mistakes have been corrected, we did not highlight them in the review.

Line 12: to write about “…technological and sensory properties of the final products” is a combination between two descriptors that in this context do not fit well. Yes, “sensory properties” is fine, but “technological properties of the final products” is not; at least it is not clear to this reviewer how the final products can have ‘technological properties’.

The word “parameters” has been inserted after the word “technological” as: “…technological parameters and sensory properties…”

Line 13: “Different amounts….were used”  (not was)

The reviewer is right, the word has been corrected.

Line 19: do not use “however” here, but write “but do not influence…”

The sentence has been corrected as the reviewer suggested.

Line 29: after “land [1]” you ought to cite the pioneering paper by Meyer-Rochow 1975, which started the ball rolling and led to all this current interest in insects as food (“Can Insects Help To Ease The Problem Of World Food Shortage?” Search 6(7), 261-262).

The reference has been added.

Line 31: what are subregions of Africa?

The word “subregions” has been removed since these subregions might change over time in Africa as well as in any parts of the world.

Lines 30 and 32: I suggest you first write “Malnutrition…” instead of ‘undernutrition’ and on line 32 continue with “other forms of food-related problems (hunger….”) or  “additional nourishment problems such as hunger….”

The reviewer is right, hunger is not a type of malnutrition, therefore we removed from the list.

Lines 35-37: very good!

Thank you for the kind comment.

Line 40: write “…or too high an energy intake.”

The sentence has been rephrased as: “High saturated fat, added sugar, salt and cholesterol content, as well as too low or too high energy intake are classified as dis-qualifying.”

Line 44: “…studies, the food industry…”

The sentence has been corrected.

First paragraph: very good!

Thank you for the kind comment.

Line 61: Check out the paper by Chakravorty et al (Int. J. Vitam. Nutr. Res., 81 (1), 2011, 1 – 14). It may be worth referring to it.

The reference has been added.

Line 71: Delete ‘However’ (you tend to use ‘however’ too often) and start “Using insects as….points of view, but industrial farming…”  I wonder if ‘commercial farming’ would not be more appropriate than ‘industrial’?

The sentence has been corrected as the reviewer suggested.

Line 76: “Western people are still disgusted…”  I think you cannot make such a sweeping statement.  Not all “Western people” are disgusted; there are many reasons why not many Westerners enjoy consuming insects and disgust is only one of several factors. Using that word is insulting  to millions of people who love insects (not necessarily to eat), but to enjoy, to keep as pets, to study, etc.

The word “disgusted” has been removed and the sentence has been rephrased.

Line 118: “at once” seems hardly the correct expression. What do you exactly mean to say? The sentence has been corrected to “For the proper homogenization, batches of 300 g dough were made.”

Line 138 and Line 171: I recently reviewed a paper in which ‘spreadability’ (how quickly and fast the dough spread) was shown to depend on the amount of insect flour in the dough and how this affected the thickness and size of the cookie. Was there no such effect in your study?

The reviewer raised an interesting question which would be an interesting point to address in future studies. Unfortunately, we did not look for such changes but cannot state that spreadability did not depend on the added amount of insect flour.

Line 159-163: The educational backgrounds of the ‘assessors’ were more or less the same (all students). If you were to predict (by extension) how the general population of Budapest might have reacted, could you speculate in Discussion what the outcome may have been?

The most up-to-date study regarding consumer acceptance of insects as food was run by our research team (Gere et al., 2017). These results suggest that Hungarian consumers are less open to insects, however, our personal experience shows that when insects are presented to consumers, they usually make a try and taste them (even if the insect is not ground).

We have included these into the discussion section.

Gere, A., Székely, G., Kovács, S., Kókai, Z., & Sipos, L. (2017). Readiness to adopt insects in Hungary: A case study. Food Quality and Preference, 59. https://doi.org/10.1016/j.foodqual.2017.02.005

Table 2, Use British spelling “Odour” and ”Flavour”

The table has been corrected.

Page 7, first paragraph: excellent.

Thank you for the kind comment.

Figures: all necessary and acceptable.

Thank you for the kind comment.

Line 392: write “…compared with the control…”  When you compare something ‘t’, you are expressing likeness, but if the task was a true comparison, it has to be ‘with’ (Example: you can compare Italy’s weather to that of New Zealand  = similar. Or you can compare Italy’s weather with that of New Zealand = seeking a true comparison).

The reviewer is right, the sentence has been corrected.

The authors may want to read and cite this interesting paper “Factors Predicting the Intention of Eating an Insect-Based Product” by Simone Mancini ,Giovanni Sogari ,Davide Menozzi ,Roberta Nuvoloni ,Beatrice Torracca ,Roberta Moruzzo and Gisella Paci: Foods 2019, 8(7), 270; https://doi.org/10.3390/foods8070270 - 19 Jul 2019, in which the authors write: “Sensory scores highlighted that participants gave “insect-labelled” samples higher scores for flavour, texture, and overall liking; nevertheless, participants indicated that they were less likely to use the “insect-labelled” bread in the future.”

This paper is interesting indeed, thank you for the suggestion!

Round 2

Reviewer 1 Report

I congratulate to Authors for the improvements made to the paper. 

Author Response

Thank you so much for the positive comments.